# Multi-Scale Attention Network for Building Extraction from High-Resolution Remote Sensing Images

**DOI:** 10.3390/s24031010

**Published:** 2024-02-04

**Authors:** Jing Chang, Xiaohui He, Panle Li, Ting Tian, Xijie Cheng, Mengjia Qiao, Tao Zhou, Beibei Zhang, Ziqian Chang, Tingwei Fan

**Affiliations:** 1School of Computer and Artificial Intelligence, Zhengzhou University, Zhengzhou 450001, China; changjing925@zzu.edu.cn (J.C.); 15639932467@163.com (T.T.); mullich@outlook.com (T.Z.); 2School of Geoscience and Technology, Zhengzhou University, Zhengzhou 450001, China; zzulpl0215@zzu.edu.cn (P.L.); chengxj@zzu.edu.cn (X.C.); qiaomjj@zzu.edu.cn (M.Q.); zhangbeibei183@gs.zzu.edu.cn (B.Z.); 13215795769@163.com (Z.C.); ftw22@gs.zzu.edu.cn (T.F.); 3Ecometeorology Joint Laboratory of Zhengzhou University and Chinese Academy of Meteorological Science, Zhengzhou 450001, China

**Keywords:** remote sensing, multi-scale feature extraction, multi-scale feature fusion, adaptive weighting

## Abstract

The precise building extraction from high-resolution remote sensing images holds significant application for urban planning, resource management, and environmental conservation. In recent years, deep neural networks (DNNs) have garnered substantial attention for their adeptness in learning and extracting features, becoming integral to building extraction methodologies and yielding noteworthy performance outcomes. Nonetheless, prevailing DNN-based models for building extraction often overlook spatial information during the feature extraction phase. Additionally, many existing models employ a simplistic and direct approach in the feature fusion stage, potentially leading to spurious target detection and the amplification of internal noise. To address these concerns, we present a multi-scale attention network (MSANet) tailored for building extraction from high-resolution remote sensing images. In our approach, we initially extracted multi-scale building feature information, leveraging the multi-scale channel attention mechanism and multi-scale spatial attention mechanism. Subsequently, we employed adaptive hierarchical weighting processes on the extracted building features. Concurrently, we introduced a gating mechanism to facilitate the effective fusion of multi-scale features. The efficacy of the proposed MSANet was evaluated using the WHU aerial image dataset and the WHU satellite image dataset. The experimental results demonstrate compelling performance metrics, with the F1 scores registering at 93.76% and 77.64% on the WHU aerial imagery dataset and WHU satellite dataset II, respectively. Furthermore, the intersection over union (IoU) values stood at 88.25% and 63.46%, surpassing benchmarks set by DeepLabV3 and GSMC.

## 1. Introduction

With the rapid evolution of remote sensing technology and its expanding spectrum of applications, remote sensing imagery has acquired a heightened significance across numerous domains, particularly in the realms of urban planning [1], land use [2], building recognition [3], and others. Remote sensing images proffer real-time, precise, and comprehensive geographical information, constituting a valuable resource for comprehending urban spatial structure and its dynamic transformations. However, the intricacies inherent in remote sensing images, compounded by the diversity in building shapes [4], dimensions [5], orientations [6], and lighting conditions [7], present substantial challenges in the realm of building information extraction [8]. Conventional methods for extracting building information from remote sensing images predominantly rely on image processing and computer vision techniques, encompassing edge detection [9], threshold segmentation [10], region growth [11], etc. These techniques, however, impose stringent prerequisites concerning image quality and lighting conditions, often yielding suboptimal results in complex environmental contexts. Particularly in the domain of high-resolution and hyperspectral remote sensing imagery, traditional methods often fall short of meeting practical demands [12]. Furthermore, traditional approaches exhibit limitations in harnessing the spatial structure information inherent in buildings [13], potentially resulting in data loss or distortion during the extraction process.

In recent years, the rapid advancement of deep learning technology has precipitated considerable attention to building extraction methodologies rooted in deep neural networks [14]. These approaches leverage the inherent learning and feature extraction capabilities of deep neural networks [15] to autonomously acquire and extract building attributes from remote sensing imagery, significantly enhancing the precision and robustness of the extraction process [16]. Notably, fully convolutional networks (FCNs) [17] have emerged as a preeminent choice within the ambit of building extraction. FCNs’ deconvolution layers facilitate the mapping of network-learned features to an output image of matching dimensions, thereby enabling automatic building extraction. Building upon the foundation of FCN, researchers have devised numerous enhanced iterations, such as Unet [18], PSPNet [19], LSPNet [20], and more, further elevating the precision and efficiency of building extraction. For example, Liang et al. [21] enriched the U-Net model by introducing additional convolutional layers and skip connections, augmenting the model’s capacity to perceive intricate building details. HE et al. [22] innovatively incorporated dilated convolution and expansion convolution layers into the U-Net architecture while introducing skip connections, substantially heightening edge perception in building extraction. WU et al. [23] integrated an expansion convolution module into the PSPNet framework, efficaciously enhancing the precision of low-rise building extraction. Meanwhile, Abolfazl et al. [24] introduced bidirectional, convolutional long short-term memory, grounded in the SegNet model, significantly bolstering the model’s performance in building extraction within complex backgrounds. These pioneering research endeavors engender novel insights into and methodologies for building extraction, thereby fortifying the groundwork for the analysis and utilization of remote sensing imagery.

Buildings exhibit multi-scale characteristics owing to their unique geometric features, varying sizes, and specific locational information. However, current methods often focus on a single scale, overlooking the wealth of information available at other scales. To overcome this limitation, researchers have endeavored to integrate multi-scale information, thereby improving the accuracy and robustness of the models. In this pursuit, Li et al. [25] introduced the Multi-Level Feature Fusion Network (MFFNet), enhancing global feature extraction through pyramid pooling. This enhancement facilitates the extraction of buildings, particularly in large and complex environments. Jiabin et al. [26] employed multi-scale depth-wise spatial pyramid pooling, reducing computational complexity while extracting a more comprehensive set of multi-scale features, thereby advancing building extraction accuracy. The integration of attention mechanisms has emerged as pivotal in deep learning, especially for tasks necessitating focused analysis of specific image regions [27]. Researchers aim to further refine the accuracy and robustness of building extraction models by incorporating attention mechanisms [28]. An illustrative instance is Ye et al.’s [29] integration of attention mechanisms into the FCN model, intensifying the focus on building-related features. Zhu et al. [30] introduced an attention-based feature enhancement module that optimizes features in both the channel and spatial dimensions, significantly elevating accuracy in building contour extraction. Additionally, Das et al. [31] proposed the RCA-Net model, leveraging spatial and channel attention to capture long-range multi-scale contextual information, resulting in precise building feature extraction. Although prior research in the field of building extraction has made certain progress, these methods often rely on relatively simple fusion techniques such as summation or concatenation. However, this approach may lead to potential issues of data redundancy, which in turn increases the risk of model overfitting or underfitting. More importantly, if a uniform weight allocation method is applied to all data during the fusion process, it may overlook the differing levels of importance among buildings of various scales within an image, as well as the spatial positional dependencies among target pixel points. This oversight can result in inaccurate target detection and even introduce significant internal noise into the final results. Therefore, to more effectively address the challenges of building extraction, it is necessary to explore more advanced fusion techniques and weight allocation strategies. Hence, guided by the concept of adaptive weighting, we introduce the MSANet, a novel building extraction network underpinned by attention mechanisms. This article’s principal contributions encompass:

(1) Innovative Multi-Scale Feature Extraction: We have formulated a Multi-scale Attention Feature Extraction (MAFE) module that seamlessly amalgamates the ASPP (Atrous Spatial Pyramid Pooling) and CBAM (Convolutional Block Attention Module) mechanisms. Notably, this design encapsulates ASPP to engender comprehensive multi-scale insights. Furthermore, it leverages channel attention and spatial attention mechanisms to delve into deeper semantic understanding at each scale. This approach also optimizes network efficiency through the strategic integration of residual units.

(2) Multi-Scale Gating Attention Fusion: Building upon the notion of adaptive weighting, we have introduced a Multi-scale Gating Attention Fusion (MGAF) module, harnessing the Multi-Head Attention mechanism. This module harnesses the Multi-Head Attention mechanism to discern positional dependencies among pixels within different-scale feature maps, subsequently hierarchically weighting them to achieve the optimal feature combination. Building upon this foundation, a gating mechanism is employed to judiciously fuse features from diverse levels, culminating in the production of high-precision building feature maps.

## 2. Multi-Scale Attention Network for Building Extraction

In this section, we aim to offer an in-depth elucidation of the precise architectural design of the Multi-scale Attention Network (MSANet). Initially, we will elucidate the overarching framework and operational methodologies of the network. Subsequently, we will delve into a comprehensive examination of the pertinent functional modules. Finally, we will introduce the loss function that has been employed in this context.

### 2.1. Overall Architecture

In the encoder section, the ResNet50 backbone network [32] was employed to proficiently extract the semantic characteristics of building structures. Within the MAFE module, we employed multi-scale channel attention and multi-scale spatial attention mechanisms, enabling adaptive concentration on pivotal channels and spatial regions within the image. This approach facilitated the precise and efficient extraction of multi-scale and multi-level information pertaining to building features. Transitioning to the decoding phase, we integrated the MGAF module, which conducts adaptive hierarchical weighted processing on multi-scale building attributes, thereby yielding a more comprehensive and enriched representation of building characteristics. Furthermore, through the astute utilization of the gating mechanism [33], we proficiently merged the low-level feature maps’ spatial details with the high-level feature maps’ semantic information. This approach enabled the in-depth exploration of multi-level and multi-scale attributes, culminating in the production of high-precision building feature maps. To address the challenge of vanishing gradients that often arises with deep neural networks, we thoughtfully incorporated skip connections between the encoder and the decoder. This design choice further safeguards the model training process’s stability. The overall network framework is shown in Figure 1.

### 2.2. MAFE Module

Compared with various remote sensing targets, buildings exhibit notable distinctions in terms of their significant scale variations, diverse geometries, intricate structures, and abundant internal intricacies [34]. Nonetheless, they frequently encounter challenges stemming from occlusion by surrounding objects or neighboring entities possessing similar textures and structures. This scenario imposes formidable obstacles in the realm of feature extraction. To tackle these challenges, we devised a novel component termed the Multi-scale Attention Feature Extraction Module, as shown in Figure 2. This module seamlessly integrates pyramid pooling, an attention mechanism, and dilated convolution, enabling the effective handling of issues such as variations in building size, shifts in perspective, and alterations in scale within remote sensing imagery. This, in turn, facilitates the more precise acquisition of multi-scale building features. Given the specific attributes of buildings, which are characterized by a limited semantic content but a profusion of intricate details, we opted to employ ResNet50 as the feature extractor for buildings. In order to retain an increased amount of detailed feature information, we selectively retained only the initial four levels of the Backbone module {C_1_, C_2_, C_3_, C_4_}, and additionally employed a convolution operation with a dilation rate of 2 at the terminal layer of each module to enhance the network’s receptive field.

We took the feature map denoted as C_4_ as our input, characterized by dimensions of H × W and a channel count of C. To circumvent the potential loss of information related to small-area clustered buildings, which might occur when employing high dilation rates, we opted for dilation rates of 6, 12, and 18 [35,36,37], coupled with a kernel size of 3 × 3. This configuration ensures the preservation of small-area building features during information propagation while simultaneously enabling the robust extraction of features from larger building structures. In pursuit of acquiring a holistic perspective on the entire feature map, we incorporated a global average pooling layer. To enhance our ability to capture object boundary information while concurrently reducing the computational complexity by reducing the number of channels, we fed the extracted feature maps from various scales into a 1 × 1 convolutional layer. This operation serves to harmonize the channel dimensions across these feature maps. Ultimately, we yield a multi-scale feature map denoted as *P* = {*P*_1_, *P*_2_, *P*_3_, *P*_4_}, which encapsulates diverse scales of extracted features.

Meanwhile, with the aim of enhancing the precision of building features and eliminating redundant information, we took the feature map *C*_4_ as our input. We initially acquired the global context information through a global average pooling operation. Subsequently, we employed a series of fully connected layers or convolution layers to derive a channel attention map, denoted as *Mc*(*C*_4_), which served to establish the significance weights for each channel within the feature map. In the final step, we performed an element-wise multiplication of the channel attention map *Mc*(*C*_4_) with the multi-scale feature map *P*, yielding a weighted multi-scale feature map denoted as *P*′. This method of information interaction and weight adjustment functions to selectively amplify features pertinent to buildings while suppressing irrelevant features, thereby elevating the perceptual and recognition capabilities of the building extraction model. This process, which augments the model’s generalization and robustness, can be formally represented by Equations (1) and (2): (1)$$P^{\prime }=M_{c}⊗\left(C_{4}\right)P$$
(2)$$M_{c}\left(C_{4}\right)=\sigma \left(MLP\left(AvgPool\left(C_{4}\right)\right)+MLP\left(MaxPool\left(C_{4}\right)\right)\right)$$

Subsequent to this stage, we took the multi-scale feature map *P*′, enriched with channel weight information, as our input and began learning the significance of each spatial position, resulting in the creation of a multi-scale spatial attention feature map. This approach took full advantage of channel weight information, enabling a more comprehensive capture of spatial intricacies and building details. It served to diminish the impact of spatially redundant features on building extraction and enhance the model’s perceptual and extraction capabilities across various building scales. Furthermore, it effectively addressed issues of within-class similarity and mitigated the influence of occlusions and noise. To achieve these objectives, we employed a combination of dilated convolution and global pooling operations for the extraction of features related to buildings of different scales. Leveraging channel adjustment technology, we generated a multi-scale feature map denoted as *F*. This was achieved through a sequence of operations applied to the feature map *P*′, including feature mapping, max pooling, average pooling, similarity calculations, normalization, and feature fusion. Collectively, these operations culminated in the creation of a spatial attention map, *Ms*(*P*′), which reflected the importance weights associated with each spatial position within the feature map. Finally, through element-wise multiplication of the spatial attention map *Ms*(*P*′) with the feature map *Q*, which contained channel weight information, we obtained a new feature map, designated as *M*. This feature map encapsulated both channel weight and spatial weight information, serving as the ultimate output. The entire process can be concisely represented by Equations (3) and (4):(3)$$F=\left\{F_{1}\right.,F_{2},F_{3},\left.F_{4}\right\}$$
(4)$$M=M_{s}\left(P^{\prime }\right)⊗F$$

### 2.3. Multi-Scale Gating Attention Fusion Module

Given the substantial influence of feature hierarchy and spatial scale selection on the efficacy of contour segmentation and the level of object fragmentation in geospatial objects, it becomes imperative to refine feature amalgamation by adjusting weight allocation at each stratum of the network architecture. This optimization endeavors to markedly diminish the ambiguity of pixel classification and facilitate the extraction of geospatial objects characterized by intricate details and well-defined contours. In response to this challenge, this paper introduces a novel module at the decoding stage, denoted as the “Multi-scale Gating Attention Fusion module (MGAF).” In its first stage, an adaptive weight adjustment strategy was implemented to derive an optimized feature combination tailored for building extraction. Subsequently, a gating mechanism was introduced to effectively blend features derived from diverse levels, yielding high-precision building feature maps. For a comprehensive overview of the MGAF’s architectural composition, please refer to Figure 3. 

The utilization of a multi-head attention mechanism [38] provides the capacity to dynamically learn significant feature relationships within an input sequence while effectively filtering out crucial information from a multitude of potential interfering factors. Recognizing that the expression of building features is intricately linked to the precise position of positive sample pixels within the image, the MGAF module leverages the multi-head attention mechanism to capture the positional dependencies between pixels in feature maps associated with buildings of various scales. Consequently, this approach assigns heightened importance to key components while mitigating interference from other ground objects and noise sources. To enhance computational efficiency and mitigate the risk of overfitting, we employed a strategy of parallel processing utilizing four independent attention heads.

In our framework, we assumed that the multi-scale feature map matrix after undergoing position encoding would be denoted as $\hat{F}\in R^{H\times W\times C}$. Subsequently, we employed a linear transformation to convert this matrix into three distinct matrices, namely, *Q*, *K*, and *V*, all sharing the same dimensions. Here, *Q* represents the query matrix, *K* stands for the key matrix, and *V* corresponds to the value matrix. To gauge the similarity between building pixels, we applied the scaled dot-product method. More specifically, we performed a scaled dot-product operation between the matrices *Q* and *K*, yielding a similarity calculation result for the building pixels. To regulate the magnitude of the dot-product result, we introduced a scaling factor denoted as “d_k_” for appropriate adjustment [39,40,41]. Finally, we employed the softmax function to normalize the attention scores, subsequently multiplying the resulting weight vector with *V*. This process resulted in a weighted computation for a single-dimensional feature map. The mathematical expression governing this process, namely, Formulas (5) and (6), is presented as follows:(5)$$Q_{i}=\hat{F}W_{i}^{Q}\;\;K_{i}={\hat{F}W}_{i}^{K}\;\;V_{i}={\hat{F}W}_{i}^{V}$$
(6)$${head}_{i}=Attention\left(Q_{i},K_{i},V_{i}\right)=Softmax\left(\frac{Q_{i}K_{i}^{T}}{\sqrt{d_{k}}}\right)V_{i}$$

In Formula (5), the matrices$\;W_{i}^{Q}$, $W_{i}^{K}$ and $W_{i}^{V}$ denote the transformation matrices associated with the initial parameters of Q, K, and V for the i-Ĝth linear transformation. In Equation (6), “headi” signifies the output result derived from the i-th attention head, where i takes values from the set {1,2,3,4}.

In this study, we employed four attention heads for simultaneous computation. To account for the interplay between different scale feature maps and to consolidate the results, a final linear transformation was applied to aggregate the multi-head feature maps, yielding the ultimate outcome. The calculation formula for this process, specifically Formula (7), is as follows:(7)$$MultiHead(Q,K,V)=Concat({head}_{1},{head}_{2},{head}_{3},{head}_{4})W^{O}$$

In Equation (7), the “MultiHead” function denotes the multi-scale composite feature matrix generated after weighted summation by the multi-head attention mechanism. The “Concat” function represents the concatenation operation, and “$W^{O}$” signifies the transformation matrix. 

In the process of image restoration at the decoding stage, to create a high-level feature map encompassing shallow spatial details while embedding semantic information from high-level features into the middle and low-level feature maps, a conventional approach is to incorporate the pixel-wise downsampled result from the previous level into the corresponding low-level features from the encoding stage for feature fusion. However, this method treats both information components equally and disregards the distinct characteristics of features across different levels. Consequently, this often leads to a fusion result contaminated by significant noise. Therefore, within the MGAF module, we introduced a gating mechanism that took inputs from the feature maps at the same level of the encoder, the multi-scale feature map from the previous level, and the feature map processed by the multi-head attention mechanism. This approach enabled the generation of high-precision building feature maps. The mathematical formula governing this fusion process is presented below in Equations (8)–(10).
(8)$$H_{n}=sigmoid(conv(C_{n}))\cdot C4\prime +C_{n}+{MH}_{n}$$
(9)$$G_{n}=sigmoid(conv(H_{n}))$$
(10)$$A_{n}=G_{n}\cdot C_{n}+(1-G_{n})\cdot {MH}_{n}$$

In the provided formula, ‘n’ denotes the current feature level (with n ∈ {1,…,l}), where ‘l’ signifies the model hierarchy. $C_{n}$ represents the encoder feature map, $H_{n}$ represents the mixed feature output, and ${\prime G}_{n}$ ∈ (0,1)’ represents the gating unit. The ‘Conv’ operation denotes a regular 1 × 1 convolution operation, and $A_{n}$ signifies the fusion result.

## 3. Experiments

In this section, we aim to validate the efficacy of our proposed method. To this end, we chose two publicly accessible datasets for testing. We provide a comprehensive account of our implementation details, including specifics about the experimental environment and parameter configurations. We also elaborate on the employed evaluation metrics and the comparative methods for benchmarking our approach. 

### 3.1. Datasets

To assess the practical effectiveness and generalization performance of our proposed MSANet method, we carried out extensive experiments using two widely recognized and publicly available datasets: the WHU aerial image dataset and the WHU satellite image dataset [42]. These datasets encompass diverse building types and sizes, as well as a range of scenarios and urban environments, facilitating a comprehensive evaluation of our model’s performance. In Figure 4, we present sample cases from these datasets, including images of buildings and their respective surroundings. This visualization aids in gaining a better understanding of the model’s application scope and its real-world effects.

The WHU Aerial Imagery Dataset, sourced from the Land Information New Zealand website, covers an area of 450 square kilometers in Christchurch, New Zealand, and contains 187,000 buildings. The dataset comprises 8189 images with resolutions of 512 × 512 pixels. The original spatial resolution of the images is 0.075 m, which has been downsampled to 0.3 m for ground resolution. In this study, the dataset was divided into 4736 images (containing a total of 21,556 buildings) for the training set and 3452 images (containing a total of 56,500 buildings) for the test set.

The WHU Satellite Dataset II consists of six adjacent satellite images covering an area of 550 square kilometers in East Asia, with a ground resolution of 2.7 m and containing 29,085 buildings. The original images were seamlessly cropped into 17,388 data samples with resolutions of 512 × 512 pixels. Among these, 13,662 images (containing 25,749 buildings) were used for model training, while the remaining 3726 images (containing 7529 buildings) were used for model testing. To accommodate memory constraints, during training, these images are further cropped into 128 × 128 pixel sizes before being input into the model.

As shown in Figure 4, compared to the WHU Aerial Imagery Dataset, the WHU Satellite Dataset II presented additional challenges for building extraction tasks due to its varied colors and diverse environments.

### 3.2. Experimental Settings

The experiments were conducted on a CentOS-Linux system with hardware specifications as follows: a CPU with 32 cores, 128 GB of RAM, and a GPU with 4 DCU. The deep learning framework used for these experiments was Tensor Flow, version 1.14.0. To accelerate model computations, we also employed the Horovod distributed framework, version 0.18.2. To train the model, we utilized the Adam optimizer, with an initial learning rate set to 0.0001 and optimizer parameters β_1 = 0.9 and β_2 = 0.999. The batch size was configured at 32, and we conducted 100 training rounds. To prevent model weights from straying away from the optimal solution, we employed a natural exponential decay parameter of 0.001 to adjust the learning rate.

### 3.3. Evaluation Metrics

In this research, we employed accuracy, recall, F1-score, and intersection over union (IoU) as the evaluation metrics for our model. Accuracy quantifies the ratio of correctly predicted positive samples to all predicted samples and is instrumental in gauging the model’s proficiency in terms of correctly identifying target categories. Recall evaluates the ratio of correctly predicted positive samples to the actual positively labeled samples, offering insights into the model’s effectiveness in correctly identifying all target instances. The F1-score represents a composite metric that factors in both accuracy and recall, providing a comprehensive assessment of the model’s overall performance. IoU measures the degree of overlap between the predicted and actual labeled regions. A higher IoU value signifies more accurate prediction results. The formulae for these metrics are as follows:(11)$$Precision=\frac{TP}{TP+FP}$$
(12)$$Recal=\frac{TP}{TP+FN}$$
(13)$$F1=\frac{Precision\times Recal}{Precision+Recal}$$
(14)$$IoU=\frac{TP}{TP+FN+FP}$$

### 3.4. Benchmark Method

We included five state-of-the-art (SOTA) semantic segmentation networks as benchmark techniques in our experiments: Unet, SegNet, ViT-V, PSPNet, and DeepLabV3.

Unet’s robustness and its capacity to retain detailed information make it highly versatile, enabling excellent performance across a wide range of application scenarios. SegNet excels in pixel-level segmentation, and its fine feature representation based on VGGNet underpins its effectiveness in various image segmentation tasks. ViT-V, with its Transformer architecture and specialized loss functions, effectively addresses long-distance dependency issues, enhancing its suitability for semantic segmentation tasks. PSPNet leverages its Pyramid Pooling module to aggregate contextual information from different regions, seamlessly embedding challenging contextual features of a scene into a pixel prediction framework, thereby elevating prediction accuracy. DeepLabV3 adopts atrous spatial pyramid pooling and a novel decoder module to achieve multi-scale feature capture and fusion, further boosting the accuracy of segmentation tasks.

In our comparative analysis, MSANet was pitted against several recent building extraction methods, which included Refined Attention Pyramid Networks (RAPNets) [28], Gate and Attention Module (GAMNet) [43], Gated Spatial Memory, and Centroid-Aware Network (GSMC) [44].

RAPNet is built upon an encoder–decoder architecture that integrates acyclic convolution, deformable convolution, attention mechanisms, and pyramid pooling modules. This amalgamation serves to augment the feature extraction capabilities of the encoding path. GAMNet, on the other hand, introduces gating and attention mechanisms to facilitate multi-level feature selection and optimize boundary details within the segmentation process. GSMC reinforces significant features and compensates for missing information through the incorporation of a gating mechanism. This approach effectively reduces the interference posed by complex backgrounds in building extraction tasks.

## 4. Results and Analysis

### 4.1. Comparison with SOTA Methods

To thoroughly assess the performance of various feature extraction algorithms for building extraction from remote sensing images, this study conducted comprehensive comparative experiments on models, including Unet, SegNet, ViT-V, PSPNet, and DeepLabV3. Specifically, we excluded the fully connected layer from the aforementioned model architectures and employed the final constitutional result from the convolutional block Cn as the output of the encoding end. Furthermore, to maintain the objectivity of the extraction results, we omitted the fifth convolutional block in the ResNet series and optimized it using a dilated convolution operation with a dilation rate of 2. The results of our quantitative comparisons are presented in Table 1 and Table 2.

In a comprehensive series of comparative experiments, our proposed MSANet model emerged as the top performer, achieving the highest IoU and F1 scores on both the WHU aerial imagery dataset and WHU satellite dataset II. This underscores the effectiveness of our approach. On the WHU aerial imagery dataset, MSANet achieved an impressive IoU score of 88.25%, surpassing the highest-scoring model by 3.82%. Similarly, in terms of the F1 metric, our model obtained a remarkable score of 93.76, outperforming the top-performing model by 2.2%. These substantial improvements highlight the ability of MSANet to more accurately identify building edges and detailed features while effectively suppressing noise and complex backgrounds in remote sensing images, leading to enhanced building extraction accuracy. On the WHU satellite dataset II, our model exhibited a strong performance, with IoU and F1 scores reaching 77.64% and 63.46%, respectively, surpassing all other models tested. This outcome underscores the generalization performance and robustness of our method when applied to different datasets. 

In Figure 5, we can clearly see that the MSANet proposed in this paper can effectively extract building targets from remote sensing images. Especially in small target recognition, MSANet shows excellent performance. To more intuitively demonstrate the extraction effect, we randomly selected four building images from the WHU aerial imagery dataset as examples for result analysis. In Scene 1, the edges and contours of the building targets are relatively accurately preserved. This indicates that MSANet has good robustness in terms of processing large targets and can effectively suppress noise interference. In Scene 2, the buildings are relatively dense, which brings challenges to target recognition. However, MSANet successfully avoids small target adhesion problems by effectively identifying the importance of small-scale features. Additionally, MSANet also detects targets that are missed, indicating its certain robustness and accuracy. In Scene 3, buildings are relatively evenly distributed, and some buildings have irregular shapes. Nevertheless, MSANet is still able to extract complete building targets and ensure clear contours and relative independence between targets. In Scene 4, there are some buildings with irregular shapes. Faced with this challenge, MSANet demonstrated superior performance. By precisely capturing the contours of buildings, MSANet avoided the emergence of voids.

For the Wuhan satellite image dataset, due to the sparse distribution of buildings and complex ground object interference, the extraction difficulty was increased. When facing terrain and vegetation interference, networks such as Unet, SegNet, ViT-V, and PSPNet all exhibit certain limitations, resulting in problems such as adjacent target adhesion and target misclassification. In contrast, DeepLabV3 performs poorly in terms of extracting small building targets, making it difficult to accurately identify and extract building details. However, MSANet exhibits significant advantages when processing such datasets. As shown in Figure 6, for closely arranged buildings, such as those that are connected or close together, MSANet can effectively maintain the independence between targets and avoid adhesion problems while accurately capturing the outlines and details of buildings. This advantage is due to MSANet’s accurate recognition and extraction capabilities for small-scale features. Through the analysis of Scene 2, Scene 3, and Scene 4 in Figure 6, we can gain a deeper understanding of MSANet’s performance. In Scene 2, although buildings were densely arranged, MSANet was able to accurately identify each building target and clearly outline it. In Scene 3, buildings were generally evenly distributed, but there were some irregularly shaped buildings. MSANet also performed well in handling such problems, accurately extracting complete building targets. In Scene 4, there were some irregularly shaped buildings that posed challenges in terms of target extraction. However, MSANet successfully avoided the appearance of holes by accurately capturing the outlines of buildings.

### 4.2. Comparison with Recent Methods

In the process of evaluating the performance of our proposed network, we conducted comparative assessments with state-of-the-art methods, which included RAPNet, GAMNet, and GSMC, on two distinct datasets. The quantitative results of these comparisons are presented in Table 3 and Table 4. The experimental outcomes highlighted that our proposed MSANet outperformed the competition on both the WHU aerial imagery dataset and WHU satellite dataset II. In all the evaluation metrics, MSANet achieved higher accuracy, recall, and F1 scores compared to the other methods. Furthermore, our method demonstrated superior capabilities in handling image details and complex backgrounds, enabling it to better capture the crucial image features. Moreover, we conducted visual comparisons that revealed our method’s proficiency in capturing both local and global image features, leading to more accurate classifications. These results collectively affirm the effectiveness of our approach.

To further understand the performance of MSANet in building segmentation tasks, we compared its performance on the WHU aerial imagery dataset with those of other advanced networks. Figure 7 visually demonstrates the effectiveness difference between MSANet and other networks in building extraction. In Scene 1, buildings were densely arranged, and many networks faced the problem of small target adhesion. However, MSANet successfully solved this problem through its ability to effectively identify small-scale features. Compared with other networks, it exhibited significant advantages in handling such dense scenes. This feature gives MSANet a significant advantage in processing highly dense scenes such as urban landscapes. In Scene 2, facing buildings with irregular shapes, the extraction effect of MSANet was significantly better than those of other networks. It can more accurately identify and extract the details of buildings, and is less susceptible to interference from internal factors of the target. Compared with RAPNet, GAMNet, and GSMC, MSANet exhibits stronger robustness in handling such challenging scenarios, effectively alleviating the problem of salt and pepper noise. For multi-story buildings in Scene 3 and 4, MSANet also demonstrated its superiority. Compared with other networks, it can more accurately capture the outlines of buildings and avoid the appearance of holes. This feature gives MSANet a significant advantage in processing complex building structures.

As shown in Figure 8, MSANet was compared with RAPNet, GAMNet, and GSMC on the Wuhan satellite image dataset. When faced with terrain and vegetation interference, other networks had different degrees of problems involving adjacent target adhesion and target misclassification, while MSANet relatively accurately preserved the edges and contours of the target, showing good robustness to noise interference. In Scene 3, the overall distribution of the buildings was relatively uniform, and some of the buildings had irregular shapes. Despite this, MSANet was still able to extract complete building targets and ensure clear contours and relative independence between the targets. Compared with other networks, it showed significant advantages in handling such scenarios. In Scene 4, for buildings that were connected or close to each other, MSANet not only effectively maintained the independence between targets and avoided adhesion problems, but also accurately captured the contours and details of the buildings. Compared with other networks, it shows stronger robustness in handling such challenging scenarios.

## 5. Conclusions

This paper introduces a novel multi-scale attention network (MSANet) designed for the extraction of buildings from high-resolution remote sensing images. In the encoding phase, the Atrous Spatial Pyramid Pooling (ASPP) technique was employed to effectively capture multi-scale information. Furthermore, attention mechanisms were integrated to facilitate the acquisition of deep semantic representations at each scale, culminating in the extraction of comprehensive and hierarchical building features. In the decoding phase, an adaptive weighting scheme and gating mechanism were implemented. This strategic approach enabled the discernment of the significance of building targets within the image while considering the spatial positional dependencies of pixel points. As a consequence, this methodology facilitated a more precise restoration of spatial details within high-level features, consequently enhancing the effectiveness of feature fusion. In substantiating the efficacy of our algorithm, we conducted rigorous experiments utilizing both the WHU aerial image dataset and the WHU satellite image dataset. Our empirical investigations revealed a pronounced superiority, as demonstrated by the attainment of F1 and IoU metrics of 92.88% and 88.25%, respectively, on the aerial dataset and 78.82% and 63.46%, correspondingly, on the satellite dataset. Notably, our approach exhibited a conspicuous outperformance when juxtaposed with established networks, including Unet, SegNet, ViT-V, PSPNet, and DeepLabV3. Moreover, in the domain of semantic segmentation tasks pertaining to architectural images, the eminence of our methodology was further underscored through meticulous comparative analyses with other contemporary methodologies, thus fortifying its position as a preeminent solution in the field. Through experimental validation, MSANet has demonstrated exceptional performance in handling buildings with irregular shapes. It accurately depicts the contours of buildings, significantly avoiding the occurrence of hollow phenomena. This stands in stark contrast to previous research efforts, such as those by Li et al. [25] and Zhu et al. [30], which often encountered issues like blurred edges or hollows when dealing with irregular shapes. Notably, in scenarios where buildings are closely adjacent, MSANet successfully addresses the challenge of small target adhesion by precisely recognizing the importance of small-scale features. This echoes the challenges faced by Das et al. [31], but MSANet offers a more efficient solution, giving it a significant advantage when dealing with highly dense scenarios like urban landscapes. Furthermore, in the face of complex background interferences from terrain and vegetation, MSANet not only maintains the independence between targets effectively, but also extracts small targets more clearly and accurately. This fully attests to MSANet’s robust capabilities in handling complex backgrounds and target diversity. Compared to earlier methods, such as those by Abolfazl et al. [24], which suffered from false detections and missed detections in similar scenarios, MSANet’s performance is more stable and outstanding.

In our future work, we plan to further optimize the network structure of MSANet to enhance its generalization ability on larger datasets. To achieve this goal, we will draw inspiration from the advanced technologies mentioned in papers such as [45,46], including more efficient attention mechanisms or more powerful decoding strategies. The application of these technologies will help to further improve the performance of MSANet and drive the development of remote sensing image analysis. Additionally, we acknowledge that MSANet still has limitations in certain aspects, such as building extraction in complex backgrounds or occlusion situations. To address these issues, we will continue to explore and investigate relevant improvement methods. At the same time, we will also focus on other remote sensing image analysis tasks, such as road extraction, water body monitoring, etc., to expand the application scope of MSANet.

## Figures and Tables

**Figure 1 sensors-24-01010-f001:**
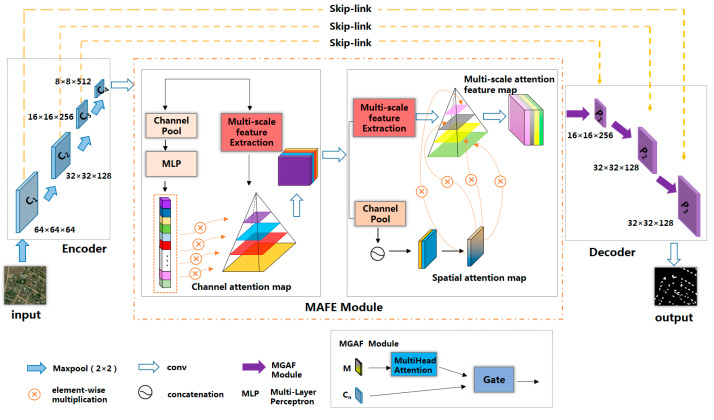
Overall architecture of the proposed network model.

**Figure 2 sensors-24-01010-f002:**
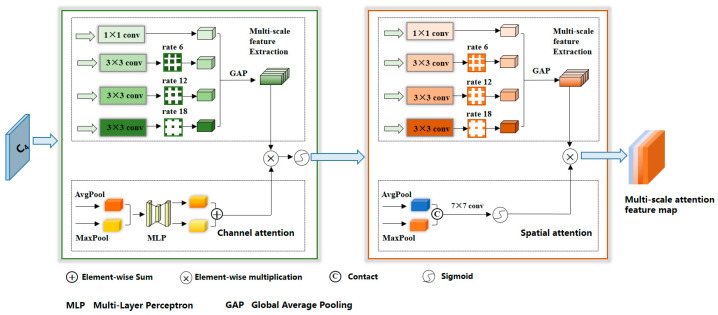
Detailed structure of the MAFE.

**Figure 3 sensors-24-01010-f003:**
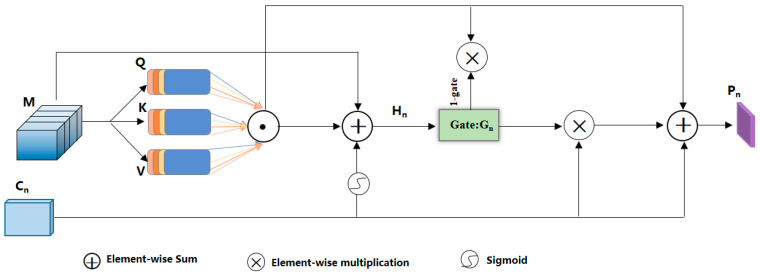
Detailed structure of the MGAF.

**Figure 4 sensors-24-01010-f004:**
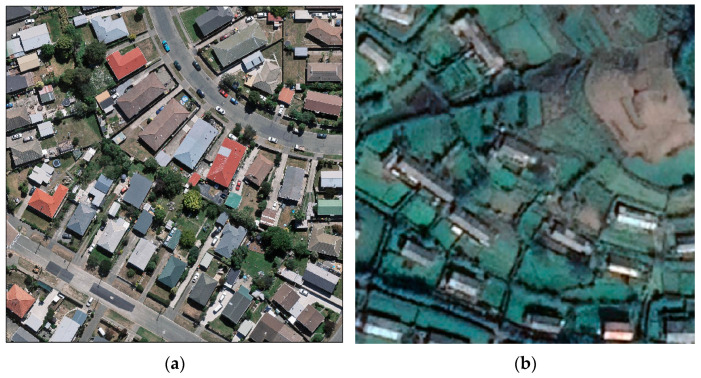
Sample images from the WHU aerial image dataset (**a**) and the WHU satellite image dataset (**b**).

**Figure 5 sensors-24-01010-f005:**
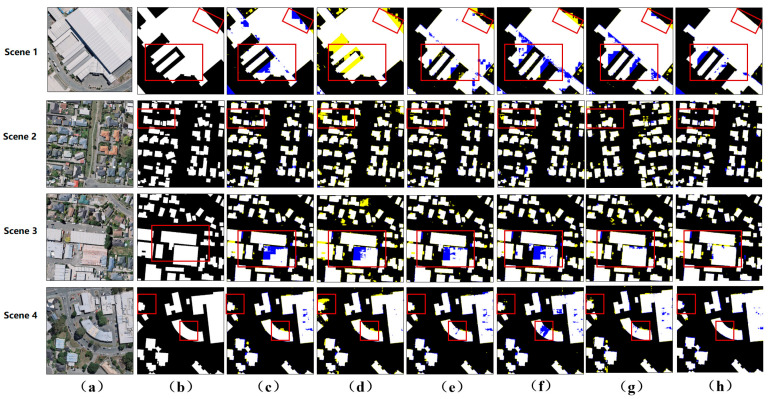
Visualization of the WHU aerial imagery dataset. (**a**) The input image. (**b**) The ground truth. (**c**) The results of Unet. (**d**) The results of SegNet. (**e**) The results of ViT-V. (**f**) The results of PSPNet. (**g**) The results of DeepLabV3. (**h**) The results of our model. The yellow and blue classification maps represent FP and FN, respectively.

**Figure 6 sensors-24-01010-f006:**
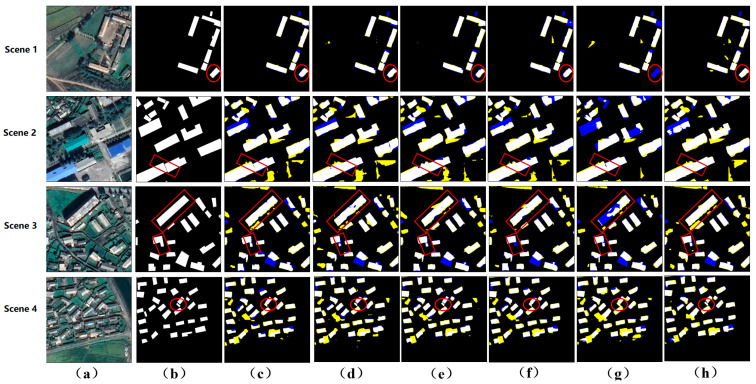
Visualization on the WHU satellite imagery dataset. (**a**) The input image. (**b**) The ground truth. (**c**) The results of Unet. (**d**) The results of SegNet. (**e**) The results of ViT-V. (**f**) The results of PSPNet. (**g**) The results of DeepLabV3. (**h**) The results of our model. The yellow and blue classification maps represent FP and FN, respectively.

**Figure 7 sensors-24-01010-f007:**
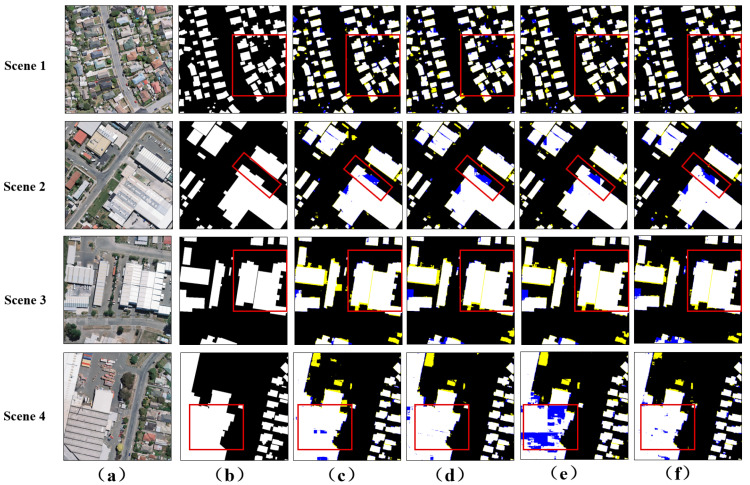
Visualization of the WHU aerial imagery dataset. (**a**) The input image. (**b**) The ground truth. (**c**) The results of RAPNet. (**d**) The results of GAMNet. (**e**) The results of GSMC. (**f**) The results of our model. The yellow and blue classification maps represent FP and FN, respectively.

**Figure 8 sensors-24-01010-f008:**
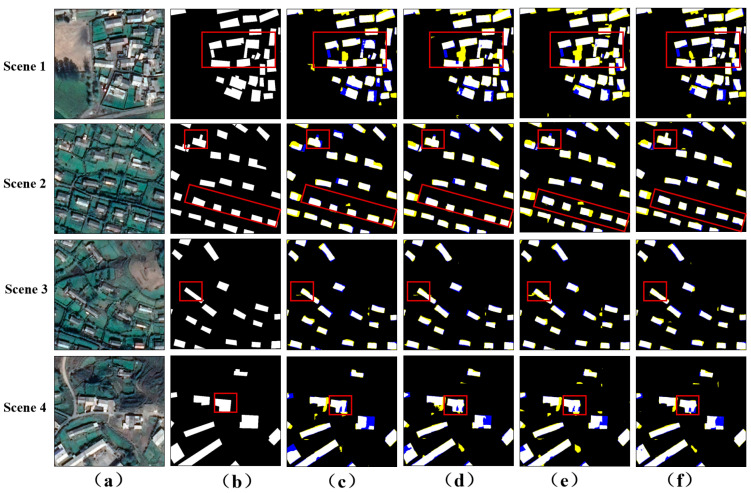
Visualization of the WHU satellite imagery dataset. (**a**) The input image. (**b**) The ground truth. (**c**) The results of RAPNet. (**d**) The results of GAMNet. (**e**) The results of GSMC. (**f**) The results of our model. The yellow and blue classification maps represent FP and FN, respectively.

**Table 1 sensors-24-01010-t001:** Comparative results from the selected models on the WHU aerial imagery dataset.

Method	Precision (%)	Recall (%)	F1 (%)	IoU (%)
Unet	88.27	94.52	91.29	83.97
SegNet	89.44	91.26	90.34	83.38
ViT-V	89.22	93.22	91.17	83.78
PSPNet	88.88	93.71	91.23	83.88
DeepLabV3	94.18	89.08	91.56	84.43
Our model	92.88	94.65	93.76	88.25

**Table 2 sensors-24-01010-t002:** Comparative results from the selected models on the WHU satellite dataset II.

Method	Precision (%)	Recall (%)	F1 (%)	IoU (%)
Unet	75.62	76.14	75.88	61.13
SegNet	71.72	71.94	71.83	53.44
ViT-V	78.44	72.70	75.46	60.59
PSPNet	71.70	76.09	73.83	58.52
DeepLabV3	76.13	73.76	74.92	59.91
Our model	78.82	76.50	77.64	63.46

**Table 3 sensors-24-01010-t003:** Comparison of state-of-the-art methods and our model on the WHU aerial imagery dataset.

Method	Precision (%)	Recall (%)	F1 (%)	IoU (%)
RAPNet	92.34	93.16	92.75	86.47
GAMNet	91.18	94.75	92.93	86.79
GSMC	91.79	94.20	92.98	86.87
Our model	92.88	94.65	93.76	88.25

**Table 4 sensors-24-01010-t004:** Comparison of state-of-the-art methods and our model on the WHU satellite dataset II.

Method	Precision (%)	Recall (%)	F1 (%)	IoU (%)
RAPNet	76.73	75.71	76.21	60.78
GAMNet	77.88	75.06	76.44	61.18
GSMC	76.63	76.43	76.53	61.98
Our model	78.82	76.50	77.64	63.46

## Data Availability

The datasets used in this study are openly available. These datasets can be found at http://gpcv.whu.edu.cn/data/ for the WHU Building Dataset.

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
