# Peer review of "Multi-Scale Attention Network for Building Extraction from High-Resolution Remote Sensing Images"

_sensors, 2024, doi:10.3390/s24031010_

Round 1

Reviewer 1 Report

Comments and Suggestions for Authors

Dear authors

The paper introduces a novel multi-scale attention network (MSANet) designed for building extraction from high-resolution remote sensing images. The integration of extraction of multi-scale building feature information and attention mechanisms in the encoding phase enhances the network's ability to capture multi-scale information and deep semantic representations. The adaptive weighting scheme and gating mechanism in the decoding phase further contribute to discerning the significance of building targets and refining spatial details, improving feature fusion effectiveness.

The empirical results, particularly the F1 and IoU metrics on both aerial and satellite datasets, demonstrate the pronounced superiority of MSANet compared to established networks such as Unet, SegNet, ViT-V, PSPNet, and DeepLabV3. The paper's meticulous comparative analyses with contemporary methodologies in the domain of semantic segmentation for architectural images underscore the eminence of the proposed methodology.

Overall, the paper is well-structured, and the methodology's effectiveness is convincingly supported by rigorous experiments and comparisons. However, to give more significance to the obtained results, I would appreciate it if the accuracy assessment and other results, if possible, could be disaggregated for data from strictly urban (city) areas with more artificial surfaces and areas with houses where there is more greenery.

Overall the paper give a valuable contribution to the field of building extraction from remote sensing images.…..

All my other comments are provided in the accompanying document.

Thank you.

Author Response

Dear Reviewer,

Greetings!

Firstly, I would like to express my deepest gratitude for the valuable time and effort you have invested in reviewing our manuscript. Each of your comments and suggestions is crucial to us, providing clear directions for improvement and significantly enhancing the quality of our work.

In response to your feedback, our team has engaged in thorough discussions and made meticulous revisions. We have striven to ensure that the revised manuscript is clearer, more precise in its presentation, and more rigorous and coherent in its logical structure. Additionally, I want to specifically acknowledge your words of encouragement, which have greatly boosted our confidence in our research efforts.

For your convenience, we have compiled the detailed revisions into an attachment and uploaded it. We hope this will facilitate a swift understanding of the improvements we have made to the manuscript.

We fully appreciate that each review process is both a rigorous test of our research and a valuable opportunity for growth. Therefore, we extend our heartfelt thanks once again for your diligent work and invaluable insights. We look forward to continuing to benefit from your guidance and expertise, propelling our research forward.

If, upon reviewing the revised manuscript, you have any further suggestions or comments, please do not hesitate to contact us. We are committed to listening carefully, remaining open to feedback, and striving for continuous improvement in pursuit of higher academic standards.

Thank you once again for your dedication and invaluable feedback! We wish you the utmost success and well-being in all your endeavors.

Sincerely,  

Jing Chang

19 January 2024

Reviewer 2 Report

Comments and Suggestions for Authors

Introduction: While several deep learning methods for building extraction have been mentioned in paragraphs 2 and 3, the explanations on lines 95-101 do not address the limitation and a rationale for new research.

Lines 120-124 are not necessary.

Figure numbers should be properly mentioned in the text. In addition, word capitalization should be revised in the figures.

Equation numbers should be mentioned in the text.

Line 173: How the dilation rates of 6, 12, and 18 were considered?

Line 251: How is the value for the scale factor defined?

Line 324 and 325: System configuration needs to be updated.

Comments on the Quality of English Language

The English language is fine. However, a few typos and editorial mistakes should be corrected.

Author Response

(The authors gave the same response as above.)
